# Anti-Inflammatory Activity of Oxyresveratrol Tetraacetate, an Ester Prodrug of Oxyresveratrol, on Lipopolysaccharide-Stimulated RAW264.7 Macrophage Cells

**DOI:** 10.3390/molecules27123922

**Published:** 2022-06-18

**Authors:** Wuttinont Thaweesest, Visarut Buranasudja, Rianthong Phumsuay, Chawanphat Muangnoi, Opa Vajragupta, Boonchoo Sritularak, Paitoon Rashatasakhon, Pornchai Rojsitthisak

**Affiliations:** 1Center of Excellence in Natural Products for Ageing and Chronic Diseases, Chulalongkorn University, Bangkok 10330, Thailand; t.tinon@hotmail.com (W.T.); rianthong_p@hotmail.com (R.P.); opa.v@chula.ac.th (O.V.); boonchoo.sr@chula.ac.th (B.S.); pornchai.r@chula.ac.th (P.R.); 2Pharmaceutical Chemistry and Natural Products Program, Faculty of Pharmaceutical Sciences, Chulalongkorn University, Bangkok 10330, Thailand; 3Department of Pharmacology and Physiology, Faculty of Pharmaceutical Sciences, Chulalongkorn University, Bangkok 10330, Thailand; 4Cell and Animal Model Unit, Institute of Nutrition, Mahidol University, Nakhon Pathom 73170, Thailand; chawanphat.mua@mahidol.ac.th; 5Molecular Probes for Imaging Research Network, Faculty of Pharmaceutical Sciences, Chulalongkorn University, Bangkok 10330, Thailand; 6Department of Pharmacognosy and Pharmaceutical Botany, Faculty of Pharmaceutical Sciences, Chulalongkorn University, Bangkok 10330, Thailand; 7Department of Chemistry, Faculty of Science, Chulalongkorn University, Bangkok 10330, Thailand; paitoon.r@chula.ac.th; 8Department of Food and Pharmaceutical Chemistry, Faculty of Pharmaceutical Sciences, Chulalongkorn University, Bangkok 10330, Thailand

**Keywords:** oxyresveratrol, Caco-2, permeation, iNOS, COX-2, MAPKs

## Abstract

Oxyresveratrol (OXY) has been reported for its anti-inflammatory activity; however, the pharmaceutical applications of this compound are limited by its physicochemical properties and poor pharmacokinetic profiles. The use of an ester prodrug is a promising strategy to overcome these obstacles. In previous researches, several carboxylate esters of OXY were synthesized and oxyresveratrol tetraacetate (OXY-TAc) was reported to possess anti-melanogenic and anti-skin-aging properties. In this study, in addition to OXY-TAc, two novel ester prodrugs of OXY, oxyresveratrol tetrapropionate (OXY-TPr), and oxyresveratrol tetrabutyrate (OXY-TBu), were synthesized. Results from the Caco-2-permeation assay suggested that synthesized ester prodrugs can improve the membrane-permeation ability of OXY. The OXY-TAc exhibited the most significant profile, then this prodrug was chosen to observe anti-inflammatory activities with lipopolysaccharide (LPS)-stimulated RAW264.7 macrophages. Our results showed that OXY-Tac significantly alleviated secretion of several pro-inflammatory mediators (nitric oxide (NO), interleukin-6 (IL-6), and tumor necrosis factor-alpha (TNF-α)), mitigated expression of enzyme-regulated inflammation (inducible nitric oxide synthase (iNOS) and cyclooxygenase-2 (COX-2)), and suppressed the MAPK cascades. Interestingly, the observed anti-inflammatory activities of OXY-TAc were more remarkable than those of its parent compound OXY. Taken together, we demonstrated that OXY-TAc improved physicochemical and pharmacokinetic profiles and enhanced the pharmacological effects of OXY. Hence, the results in the present study would strongly support the clinical utilities of OXY-TAc for the treatment of inflammation-related disorders.

## 1. Introduction

Inflammation is a biologically defensive mechanism that protects the host from deleterious xenobiotics, such as infectious pathogens, toxicants, pollutants, and irradiation. Aberrant control of inflammatory responses can cause several deleterious disorders, such as rheumatoid arthritis, atherosclerosis, diabetes, hepatitis, cancer, and neurodegenerative diseases [1]. Non-steroidal anti-inflammatory drugs (NSAIDs) are widely used medicines for the treatment of inflammation-related diseases; however, the major limitation of these drugs is their undesired adverse events on the gastrointestinal system [2]. Hence, an alternative therapeutic compound is required to combat these disorders and improve patients’ quality of life.

The macrophage is an essential immune cell that plays a pivotal role in the process of inflammation. When a macrophage is exposed to inflammatory stimuli, the mitogen-activated protein kinase (MAPKs) cascade is triggered. Activation of this intracellular signaling pathway can enhance the production of various pro-inflammatory mediators, such as nitric oxide (NO), prostaglandin-E2 (PGE2), pro-inflammatory cytokine (e.g., interleukin-6 (IL-6), and tumor necrosis factor-alpha (TNF-α). These generated mediators are essential contributors to the regulation of the inflammatory response [3,4]. Hence, compounds that can suppress activation of the MAPK pathway and inhibit the release of pro-inflammatory mediators could be potential candidates for development as anti-inflammatory agents.

Oxyresveratrol (OXY, trans-2′,3,4′,5-tetramethoxystilbene; Figure 1A) is a naturally occurring polyphenolic stilbene found in the heartwood of the Thai traditional medicinal plant *Artocarpus lakoocha* Buch.-Ham. [5]. OXY possesses a variety of pharmacological activities, including anti-viral, anti-hyperglycemia, antioxidant, and anti-inflammatory effects. Previous studies have demonstrated that OXY has the potential to be clinically developed for the treatment of inflammation-related diseases. RAW 264.7 macrophages are common in vitro model for screening anti-inflammatory candidates. Supplementation with OXY inhibited activation of MAPK cascades and suppressed secretion of pro-inflammatory mediators from RAW 264.7 following exposure to an inflammatory stimulus in the form of lipopolysaccharide (LPS) [6,7,8,9]. Similar protective results were observed from many preclinical models of inflammation-related diseases, e.g., spinal cord injury [10], neurodegenerative disorder [11], gastric ulcer [12], and colitis [13].

Although OXY has excellent anti-inflammatory effects, the major obstacles to the pharmaceutical use of this polyphenolic stilbene are mainly due to its physicochemical and pharmacokinetic properties. The unfavorable characteristics of OXY include (i) low water solubility, (ii) chemical instability, and (iii) poor oral bioavailability [14,15,16,17]. Thus, a novel strategy to overcome these limitations of OXY is necessary. An ester prodrug is an attractive approach for improving the physicochemical and pharmacokinetic profiles of this interesting compound. A prodrug is a chemical derivative of the parent drug that must be converted to a pharmacologically active drug through enzymatic biotransformation in the body [18,19]. In this current study, we synthesized a series of OXY ester prodrugs, including tetraacetyl oxyresveratrol (OXY-TAc), oxyresveratrol tetrapropionate (OXY-TPr), and oxyresveratrol tetrabutyrate (OXY-TBu). Using Caco-2 cells as surrogates to predict intestinal permeability, we found that ester prodrugs can improve the membrane-permeation ability of OXY. OXY-TAc has the strongest ability to permeate across the intestinal membrane. OXY-TAc was therefore chosen for evaluation of its anti-inflammatory effects, using RAW 264.7 macrophages as models and LPS as an inflammatory inducer. We clearly demonstrated that OXY-Tac significantly inhibited the secretion of several pro-inflammatory mediators and suppressed the MAPK pathway in LPS-induced macrophages. Moreover, the observed anti-inflammatory activities of OXY-TAc were even more significant than its parent compound. Our results in this present study strongly support the potential application of OXY-TAc for the treatment of inflammation-related diseases.

## 2. Materials and Methods

### 2.1. Materials

Commercially available OXY (99.25% purity) was purchased from B&P Ingredients Ltd. (Shanghai, China). All chemicals were purchased from Merck Millipore (Darmstadt, Germany), Sigma-Aldrich (St. Louis, MO, USA), and Fluka (Buchs, Switzerland). All column chromatography was performed using silica gel 60 (70–230 mesh) purchased from Merck Millipore (Darmstadt, Germany). ^1^H- and ^13^C-NMR spectra were recorded on a Bruker NMR spectrometer operating at 300 MHz for ^1^H and 75 MHz for ^13^C (Bruker Company, Fällanden, Switzerland). Mass spectra were measured by high-resolution electrospray ionization mass spectra (HR-ESI-MS) (Bruker Company, Billerica, MA, USA). HPLC analysis was performed using an Agilent series 1290 UHPLC (Agilent Technologies, Santa Clara, CA, USA).

### 2.2. Synthesis and Structure Elucidation of OXY Ester Prodrugs

OXY ester prodrugs were synthesized by a slight modification of the previously reported procedure [20]. OXY (1.0 g, 4.0 mmol) was dissolved in a mixture of pyridine (4 mL) and acid anhydride (16.8 mmol) at room temperature. Acetic anhydride, propionic anhydride, and butyric anhydride were used to synthesize OXY-TAc, oxyresveratrol tetrapropionate (OXY-TPr), and oxyresveratrol tetrabutyrate (OXY-TBu), respectively. The reaction mixture was stirred under a nitrogen atmosphere for 24 h. Then, the reaction mixture was quenched with saturated NaHCO_3_ and was extracted with ethyl acetate. The pooled ethyl acetate layer was washed with water and brine, dried over Na_2_SO_4_, filtered, and concentrated under a high vacuum to obtain the product. The products were purified by flash chromatography on silica gel and eluted with 50% acetone in hexane evaporation of the solvent afforded the product.

The white solid of OXY-TAc was obtained in a 26% yield. The final yield was comparable with previous reports with similar reactions [21,22,23,24] (Appendix A). ^1^H-NMR (300 MHz, DMSO-*d*_6_), δ (ppm): 7.85 (d, *J* = 8.6 Hz, 1H), 7.35 (d, *J* = 2.1 Hz, 2H), 7.27 (d, *J* = 16.4 Hz, 1H), 7.20 (d, *J* = 16.6 Hz, 1H), 7.12 (dd, *J* = 8.6, 2.4 Hz, 1H), 7.06 (d, *J* = 2.3 Hz, 1H), 6.94 (t, *J* = 2.1 Hz, 1H), 2.39 (s, 3H), 2.29 (d, *J* = 3.2 Hz, 9H). ^13^C-NMR (75 MHz, DMSO-*d*_6_), δ (ppm): 172.4, 154.1, 150.4, 141.6, 129.7, 127.6, 124.8, 120.5, 118.0, 117.5, 116.0, 21.3. High-resolution MS (HRMS) calculated for C_22_H_20_O_8_Na (M + Na)^+^: 435.1056; found: 435.1041.

The white solid of OXY-TPr was obtained in a 32% yield. ^1^H-NMR (300 MHz, DMSO-*d*_6_), δ (ppm): 7.84 (d, *J* = 8.8 Hz, 1H), 7.32 (d, *J* = 2.2 Hz, 2H), 7.26 (d, *J* = 16.5 Hz, 1H), 7.19 (d, *J* = 16.4 Hz, 1H), 7.12 (dd, *J* = 8.6, 2.4 Hz, 1H), 7.05 (d, *J* = 2.4 Hz, 1H), 6.94 (t, *J* = 2.1 Hz, 1H), 2.74 (q, *J* = 7.5 Hz, 2H), 2.62 (qd, *J* = 7.5, 2.6 Hz, 6H), 1.15 (tt, *J* = 7.5, 3.6 Hz, 12H). ^13^C-NMR (75 MHz, DMSO-*d*_6_), δ (ppm): 172.9, 157.9, 151.7, 150.2, 149.5, 139.5, 129.6, 127.6, 127.3, 123.3, 120.4, 117.9, 117.4, 115.9, 27.4, 9.3. HRMS calculated for C_26_H_28_O_8_Na (M + Na)^+^: 491.1682; found: 491.1701.

The white solid of OXY-TPr was obtained in a 31% yield. ^1^H-NMR (300 MHz, DMSO-*d*_6_), δ (ppm): 7.85 (d, *J* = 8.7 Hz, 1H), 7.32–7.24 (d, *J* = 16.5 Hz, 1H), 7.28 (d, *J* = 17.1 Hz, 2H), 7.16 (d, *J* = 17.1 Hz, 1H), 7.14–7.08 (m, 1H), 7.04 (d, *J* = 2.4 Hz, 1H), 6.93 (t, *J* = 2.0 Hz, 1H), 2.69 (t, *J* = 7.2 Hz, 2H), 2.58 (td, *J* = 7.3, 2.1 Hz, 6H), 1.69 (dd, *J* = 7.0, 2.5 Hz, 8H), 0.98 (td, *J* = 7.4, 3.1 Hz, 12H). ^13^C-NMR (75 MHz, DMSO-*d*_6_), δ (ppm): 172.0, 171.9, 151.7, 150.8, 148.8, 139.5, 129.6, 127.6, 127.3, 123.2, 120.5, 117.8, 117.4, 115.9, 35.7, 18.4, 18.3, 13.8. HRMS calculated for C_30_H_36_O_8_Na (M + Na)^+^: 547.2308; found: 547.2315.

### 2.3. Cell Culture Conditions

The human epithelial colorectal adenocarcinoma cell line (Caco-2; HTB37) and murine macrophage cell line (RAW264.7; TIB71) were purchased from the American Type Culture Collection (Rockville, MD, USA). The Caco-2 cells (passage 25–35) were cultured in complete Dulbecco’s Modified Eagle Medium (cDMEM; DMEM supplemented with 15% fetal bovine serum (FBS), 1% nonessential amino acid (*v*/*v*), 1% l-glutamine (*v*/*v*), 0.2% fungizone (*v*/*v*), and 1% penicillin and streptomycin (*v*/*v*). The RAW264.7 cells (passage 5–15) were cultured in cDMEM (DMEM containing 10% FBS and 1% penicillin and streptomycin (*v*/*v*). Cells were maintained under a humidified air atmosphere containing 5% CO_2_ at 37 °C. Cell culture medium and supplements were obtained from ThermoFisher Scientific (Waltham, MA, USA).

### 2.4. Determination of a Non-Cytotoxic Concentration of OXY Ester Prodrugs on Caco-2 Cells

The cytotoxicity of OXY and three OXY ester prodrugs to Caco-2 cells was determined using the 3-(4,5-dimethylthiazol-2-yl)-2,5-diphenyltetrazolium bromide (MTT) assay [25]. The Caco-2 cells were plated into 96-well microplates (1 × 10^4^ cells/well) and cultured for 24 h. After incubation, cells were incubated with 1, 10, 25, 50, and 100 µM of OXY or each OXY ester prodrug for 24 h. A DMSO at 0.5% was used as a vehicle control. Following treatment, cells were incubated in darkness with MTT solution (20 µL; 5 mg/mL) for 4 h. The generated formazan crystals were subsequently dissolved by DMSO. The optical density of formazan solution was monitored at 540 nm by a microplate spectrometer (SPECTROstar, BMG LABTECH, Ortenberg, Germany). The results were expressed as the percentage of cell viability. The non-toxic concentrations of OXY and OXY ester prodrugs were chosen for further evaluation of membrane permeation.

### 2.5. Determination of Transport across Caco-2 Monolayers of OXY Ester Prodrugs

Caco-2 cells were seeded at a density of 4 × 10^4^ cells/well in trans-well inserts of 6-well plates (diameter 24 mm, pore size 0.4 µm; ThinCerts™-TC Einsatze, Greiner Bio-One, St. Gallen, Switzerland). Cells were cultured in 2 mL of cDMEM, and the basolateral part was added with 2 mL of cDMEM. The spent medium was replaced with a fresh complete medium every other day. After confluency, the serum content of the cDMEM was switched from 15% FBS to 7.5% FBS. The experiments were performed 21–24 days after cells reached confluence with a transepithelial electrical resistance (TEER) 500 Ω·cm^2^ [26]. Caco-2 monolayers were washed with the serum-free medium prior to incubation with OXY or OXY ester prodrug at a nontoxic concentration (50 µM in serum-free medium; 2 mL). The basolateral compartment was filled with 2 mL of phenol red-free and serum-free medium. After the sample treatment, the medium in the basolateral compartment was collected at indicated time points (15–240 min). The fractions collected at different time intervals were designated as bioavailable fractions (BF). The collected BF was blanked with N_2_ and kept at −80 °C for further studies. The P_app_ (cm/s) of OXY and OXY ester prodrug were calculated according to the following equation (Equation (1)): (1)$$P_{\mathrm{app}}=\frac{\mathrm{dQ}}{\mathrm{dt}}\times \frac{V}{A\times C_{0}}$$
where dQ/dt = the rate of permeation (µM/s); V = volume of basolateral chamber (2 cm^3^); A = surface area of insert (4.524 cm^2^); and C_0_ = initial concentration (µM) of samples. The total percentage of transport was calculated by dividing the cumulative amount of molecules transported with the initial loading concentrations.

### 2.6. HPLC Analysis of OXY in Bioavailable Fractions

Briefly, 2 mL of each sample from the transport experiment was transferred to a 15 mL centrifuge tube and mixed with an equal volume of 0.1 M potassium phosphate buffer (pH 7.1) containing 100 U of ß-glucuronidase and 0.1 U of sulfatase. The mixture was placed in a shaking water bath at 37 °C and incubated for 3 h. After that, 1 mL of 1% sodium dodecyl sulfate (SDS) in ethanol was added to the mixture and subsequently vortexed for 1 min. The mixture was extracted twice with ethyl acetate at a ratio of 1:1, sonicated for 10 min, vortexed for 2 min, and centrifuged (ROTINA380R, Tuttlingen, Germany) at 4000× *g* for 10 min. The ethyl acetate layer was combined into a 10-mL vial. The combined layers were blow-dried with nitrogen gas, and the residue was reconstituted with a mobile phase prior to HPLC analysis. Experiments were performed in four replicates.

HPLC analysis was performed using an Agilent series 1290 UHPLC (Agilent Technologies, Santa Clara, CA, USA) liquid chromatograph equipped with a binary pump, autosampler, thermostat column compartment, degasser, and diode array detector (DAD). Chromatographic separation was performed on a HALO C18 column (4.6 × 100 mm, 5 µm) and maintained at 25 °C. The injection volume of each sample was 5 µL. The mobile phases consisted of 0.1% formic acid as mobile phase A and methanol as mobile phase B with a flow rate of 1 mL/min. The gradient sequence was as follows: 40–46% B from 0 to 5 min, 46–48% B from 5 to 10 min, 48–48% B from 10 to 17 min, 48–52% B from 17 to 18 min, and 52–52% B from 18 to 23 min. The detection of OXY was performed at 320 nm.

### 2.7. Preparation of BF-OXY-TAc

Caco-2 cells were seeded into apical compartments at a density of 2.5 × 10^4^ cells/well and grown on trans-well inserts of a 6-well plate (ThinCertsTM-TC Einsatze, Greiner Bio-One, St. Gallen, Switzerland). A DMEM without phenol red (2 mL) was added into the basolateral compartment. The Caco-2 cells were cultured at 37 °C for an additional 21–24 days. The confluence of the cells was reached when the amount of FBS was reduced to approximately 7.5%. After 21–24 days of culture, the monolayer was generated with a TEER value of more than 500 Ω/cm^2^. Cells were then incubated with OXY or OXY-TAc at a non-toxic concentration (50 µM; 2 mL) for 4 h at 37 °C. In non-treated controls, serum-free medium was used and the permeate was considered as a blank BF. The collected BF was blanked with N_2_ and kept at −80 °C for intracellular ROS scavenging and in vitro anti-inflammatory activity assay.

### 2.8. Determination of a Non-Cytotoxicity of BF-OXY-TAc on RAW264.7 Cells

The cytotoxicity of BF-OXY and BF-OXY-TAc to RAW264.7 cells was observed with an MTT assay. The RAW264.7 macrophage cells were seeded in 96-well microplates (1 × 10^4^ cells/well) and cultured for 24 h. Following incubation, cells were treated with BF-OXY or BF-OXY-TAc for 24 h. A DMSO at 0.5% was used as a vehicle control. Following treatments, an MTT assay was conducted to evaluate cell viability using a similar protocol to that of the Caco-2-cell experiment.

### 2.9. Determination of Levels of Oxidative Stress

Levels of oxidative stress were evaluated by monitoring fluorescent signals generated by the oxidized DCFH-DA (Sigma-Aldrich, St. Louis, MO, USA). Non-fluorescent DCFH-DA was diffused into cells and deacetylated by cellular esterase to form DCFH, which can be rapidly oxidized to the fluorescent DCF by oxidative stress [27]. RAW264.7 cells were plated into 96-well black plates and cultured with cDMEM for 24 h. Cells were washed with phenol red-free and serum-free medium and pre-treated with BF-OXY or BF-OXY-TAc for 1 h. Then, cells were induced with or without LPS (1 µg/mL) for 24 h. Cells were washed, and 5 µM of DCFH-DA in a serum-free medium was added for 30 min. Cells were again washed with cold phosphate-buffered saline (PBS), and 200 µL of cold PBS was added prior to fluorescent measurement. The intracellular fluorescent signal of DCF was recorded at 485 nm (excitation) and 530 nm (emission) using a microplate reader (CLARIOstar, BMG LABTECH, Ortenberg, Germany). Results were presented as the percentage of reactive oxygen species (ROS) production.

### 2.10. Determination of NO, IL-6, and TNF-α Productions

RAW264.7 cells were seeded in 6-well plates (1.0 × 10^6^ cells/well) and cultured for 24 h. Cells were pre-treated with BF-OXY or BF-OXY-TAc for 1 h. Then, cells were induced with or without LPS (1 µg/mL) for 24 h. Cells with no BF treatment and LPS activation were used as the control group. The culture medium was collected for the measurement of released levels of nitrite, IL-6, and TNF-α.

The nitrite concentration in the culture medium as NO level was analyzed by the Griess reagent [28]. Briefly, 100 µL of BF-OXY or BF-OXY-TAc were incubated with 50 µL of 0.1% N-(1-naphthyl) ethylenediamine dihydrochloride (NED) and 50 µL of 1% sulfanilamide in 5% phosphoric acid at room temperature. Nitrite concentration was calculated by comparing it with the absorbance of the standard solutions of sodium nitrite prepared in a serum-free medium at 520 nm.

The levels of released IL-6 and TNF-α were investigated with an enzyme-linked immunosorbent assay (ELISA) kit following the manufacturer’s manuals (BioLegend, San Diego, CA, USA). The levels of IL-6 and TNF-α in each sample were further estimated from their respective standard curves. Results were presented as the concentration of NO, IL-6, and TNF-α.

### 2.11. Western Blot Analysis

After the indicated treatments, the cells were washed with ice-cold PBS and lysed with ice-cold RIPA lysis buffer (Cell Signaling Technology, Boston, MA, USA) containing a proteinase inhibitor (Roche Applied Science, Mannheim, Germany) and a phosphatase inhibitor (Roche Applied Science, Mannheim, Germany). The protein concentration was estimated with the BCA protein assay. Equal amounts (20 µg) of the protein samples were separated on 10% sodium dodecyl sulfate-polyacrylamide gel electrophoresis and transferred into a nitrocellulose membrane (GE Healthcare Life Sciences, Buckinghamshire, UK). The membrane was blocked with 5% skim milk in TBST (20 mM Tris-HCl, 140 mM NaCl, 0.1% (*v*/*v*) Tween-20) and incubated with primary antibody against iNOS, COX-2, phospho-p38, p38, phospho-ERK, ERK, phospho-JNK, JNK, (1:1000), or β-actin (1:20,000) (Cell Signaling Technology, Danvers, MA, USA) at 4 °C. After 24 h, the membrane was washed with TBST and incubated with the HRP-conjugated anti-rabbit IgG secondary antibody at room temperature for 90 min. The signal was developed using a chemiluminescent western blot reagent (Bio-Rad, Hercules, CA, USA). The density of target bands was quantified by the Image J software (National Institutes of Health, Bethesda, MD, USA). Experiments were performed in four replicates. The results were expressed as a relative ratio of the specific proteins and β-actin or total forms of MAPK.

### 2.12. Statistical Analysis

The results are presented as mean ± standard deviation (SD). One-way ANOVA was performed using Dunnett’s multiple comparison post hoc test to compare all groups to controls, while Tukey’s multiple comparison post hoc test was used for comparing multiple groups. The statistical methods used for each experiment are detailed in figure legends. The *p* < 0.05 values were considered statistically significant. Statistical analyses were performed with GraphPad Prism Software, version 8.0 (San Diego, CA, USA).

## 3. Results

### 3.1. Effects of OXY Ester Prodrugs on Cell Viability in Caco-2 Cells

The cytotoxic effects of OXY and three OXY ester prodrugs on Caco-2 cells were comprehensively investigated, as shown in Figure 2. The cell viability of Caco-2 cells following treatments with OXY and OXY ester prodrugs at a concentration range of 1–100 μM was evaluated using the MTT approach. The viability of Caco-2 cells was not affected by OXY and three OXY ester prodrugs when the concentrations of OXY and three OXY ester prodrugs were not greater than 50 μM compared to 0.5% DMSO as the control. This result suggests further investigation to proceed with OXY and their ester prodrugs at concentrations of 50 μM in the subsequent permeation experiments.

### 3.2. Evaluation of OXY Ester Prodrug Transport across Caco-2 Monolayers

The Caco-2 cells are often used to assess the epithelial permeation ability of compounds because they can form a monolayer with some components of the gut epithelium [29,30,31]. In this study, the determination of TEER value and phenol red analysis were used to investigate the monolayer integrity before and during the transport experiment [32]. The DMEM with phenol red was used as a medium on the apical side, while the DMEM without phenol red was added on the basolateral side to maintain pH and osmotic balance in the medium. Phenol red in DMEM was used as a leakage indicator of monolayer integrity. In this experiment, TEER values did not change, and phenol red was not detected on the basolateral side, thus indicating that the monolayers retained good integrity during the permeation of OXY and OXY ester prodrugs. The time profiles of OXY derived from OXY ester prodrugs transport across the Caco-2 monolayers are depicted in Figure 3. The transport profiles of OXY and OXY ester prodrugs across cell monolayers at different time points showed a consistent trend, wherein the transport accumulation of ester prodrugs increased with time. In the basolateral compartment, the amounts of OXY were detected in the samples of all ester prodrugs, but the ester prodrugs were not detected. The remaining amount of OXY at 4 h in the BF of the OXY group was only 2.71 µM. Although OXY ester prodrugs demonstrated a similar transport profile to OXY, the amount of free OXY found in the BF of OXY ester prodrugs was significantly different. At 240 min, the amount of OXY in BF-OXY-TAc, BF-OXY-TPr, and BF-OXY-TBu reached a maximum of 7.49, 4.82, and 3.51 µM, respectively. The transport of OXY across Caco-2 monolayers showed a P_app_ value of 1.44 (±0.17) × 10^−6^ cm/s. The OXY-TAc, OXY-TPr, and OXY-TBu showed P_app_ values of 4.25 (±0.43) × 10^−6^, 2.67 (±0.19) × 10^−6^, and 2.17 (±0.16) × 10^−6^ cm/s, respectively. All ester prodrugs were of higher permeation than the OXY. Enhanced permeation ability was achieved with most OXY ester prodrugs investigated in this study. The OXY-TAc showed a maximum improvement in permeation ability among all ester prodrugs. Transport of OXY-TAc was 2.8-fold higher than that of the OXY after 4 h. Permeation increased 1.8-fold and 1.3-fold in OXY-TPr and OXY-TBu, respectively (Figure 3). Transport of OXY-TAc was significantly greater than the values of OXY-TPr, and OXY-TBu, suggesting that the OXY-TAc had greater permeation than the OXY-TPr and OXY-TBu. Thus, the OXY-TAc was further subjected to anti-inflammatory activity tests.

### 3.3. Effects of BF-OXY-TAc on Intracellular ROS Production

Oxidative stress is known to be important in the activation of MAPK- and NF-κB-signaling pathways in response to LPS. Overproduction of ROS can result in injury issues that might initiate the inflammatory process. To investigate the effect of BF-OXY-Tac on LPS-induced oxidative stress, we determined the oxidative status of RAW264.7 cells using the fluorescent probe DCFH-DA. The cells were first pre-treated with BF-OXY or BF-OXY-Tac for 1 h, then selectively exposed to LPS for 24 h. After that, the cells were incubated for 30 min with the fluorescent probe DCFH-DA. The amount of DCFH-DA oxidized by intracellular ROS was determined using a microplate reader to measure the fluorescence intensity of oxidized DCFH-DA, which is proportional to levels of oxidative stress. When macrophages were activated with LPS, the level of oxidative stress was significantly higher than in the control group. Pre-treatment with BF-OXY or BF-OXY-Tac significantly reduced the formation of oxidative stress in LPS-induced RAW264.7 macrophage cells, as shown in Figure 4B. Surprisingly, the inhibitory activity against oxidative stress of BF-OXY-TAc was approximately 2.5 times more effective than BF-OXY. The MTT assay was used to assess the cytotoxic effects of BF-OXY-TAc in combination with LPS. The results showed that treating RAW264.7 cells with BF-OXY-TAc and LPS generated no cytotoxicity, implying that the decrease in oxidative stress from LPS-stimulated RAW264.7 macrophage cells was not caused by a reduction in the cell population (Figure 4A).

### 3.4. Evaluation of NO Production

To evaluate the anti-inflammatory activities of BF-OXY-TAc, the effect of BF-OXY-TAc on NO formation in LPS-treated macrophages was investigated. When cells were exposed to LPS for 24 h, NO production increased significantly. However, pre-treatment with BF-OXY and BF-OXY-TAc for 1 h significantly reduced NO production. When BF-OXY and BF-OXY-TAc were compared to the LPS control group, there were 28% and 67% reductions in LPS-induced NO production, respectively (Figure 4C).

Furthermore, treatment with only BF-OXY or BF-OXY-TAc did not affect NO formation in RAW264.7 cells. As a result, treatment with BF-OXY or BF-OXY-TAc with LPS did not cause cytotoxic effects on RAW264.7 cells, indicating that the reduction in NO formation from LPS-induced macrophages was not due to an alteration in cell viability (Figure 4A). The NO production of BF-OXY-TAc was approximately 2.4 times more effective than BF-OXY. These findings highlight that BF-OXY-TAc had an anti-inflammatory effect by reducing NO formation in LPS-induced macrophages at significantly greater magnitudes than BF-OXY.

### 3.5. Evaluation of IL-6 and TNF-α Production

Both IL-6 and TNF-α are essential pro-inflammatory cytokines, either of which can serve as an indicator of inflammation. To confirm the effect of BF-OXY-TAc on anti-inflammatory activity, we also observed the inhibitory effect of BF-OXY-TAc on proinflammatory cytokine secretion by measuring IL-6 and TNF-α levels in LPS-induced macrophages. Exposure of the cells to LPS increased IL-6 and TNF-α secretion 11.3-fold and 18.3-fold, respectively, compared to the untreated control, as determined by the ELISA method (Figure 4D,E). In the absence of LPS stimulation, treatment with BF-OXY or BF-OXY-TAc alone did not alter the release of IL-6 and TNF-α from macrophages. Interestingly, pre-treatment with BF-OXY and BF-OXY-TAc suppressed the production of pro-inflammatory cytokines IL-6 and TNF-α compared to cells treated with LPS alone. Supplementation with BF-OXY considerably reduced the levels of IL-6 and TNF-α by 31% and 40%, respectively, as compared to the LPS-induced macrophages. Pre-treatment with BF-OXY-TAc significantly reduced the secretion of IL-6 and TNF-α by 62% and 72%, respectively, as compared to the LPS-induced cells. These data indicated that BF-OXY-TAc could markedly inhibit the inflammatory response in LPS-stimulated RAW264.7 cells to a greater degree than BF-OXY.

### 3.6. Evaluation of the Protein Expressions of iNOS and COX-2

Western blot analysis was conducted to investigate the effects of BF-OXY-TAc on the protein expression of iNOS and COX-2 in LPS-induced RAW264.7 cells. The levels of iNOS and COX-2 were significantly upregulated, increasing about 3.9-fold and 4.6-fold, respectively, in LPS-induced macrophages (Figure 5B,C). In the absence of LPS stimulation, treatment with BF-OXY or BF-OXY-TAc alone did not interfere with the expression of these inflammatory markers. Supplementation with BF-OXY significantly reduced the levels of iNOS and COX-2 expression by 29% and 22%, respectively, as compared to the LPS- induced cells (Figure 5B,C). In contrast, pre-treatment with BF-OXY-TAc significantly suppressed the levels of iNOS and COX-2 by 61% and 54%, respectively, as compared to the LPS-induced cells (Figure 5B,C). Therefore, it can be concluded that BF-OXY-TAc exerts an an-inflammatory effect by decreasing iNOS and COX-2 expression in LPS-induced RAW264.7 cells to a greater degree than BF-OXY.

### 3.7. Evaluation of MAPK Activation

LPS also activates MAPK to regulate the synthesis and secretion of inflammatory mediators in macrophages and many other cell types. Therefore, we investigated the influence of BF-OXY-TAc on the LPS-induced activation of MAPKs such as p38, JNK, and ERK. As depicted in Figure 5D–F, stimulation with LPS significantly enhanced the expression of p-p38, p-ERK, and p-JNK, increasing it approximately 3.6-, 3.7-, and 3.2-fold, respectively, in comparison with the non-treated macrophages. Without LPS induction, BF-OXY or BF-OXY-TAc alone did not interfere with the levels of p-p38, p-ERK, and p-JNK in macrophages (Figure 5D–F). Supplementation with BF-OXY and BF-OXY-TAc significantly attenuated the LPS-induced activation of all three MAPK in RAW264.7 cells. Pre-treatment of cells with BF-OXY significantly reduced the levels of p-p38, p-ERK, and p-JNK by 40%, 20%, and 26%, respectively, as compared to the LPS-triggered macrophages (Figure 5D–F). Pre-treatment with BF-OXY-TAc significantly attenuated the expression of p-ERK, p-JNK, and p-p38 by 59%, 62%, and 64%, respectively, as compared to the LPS-induced macrophages. Therefore, it can be concluded that BF-OXY and BF-OXY-TAc exert protective effects against inflammation via modulation of p-p38, p-ERK, and p-JNK in the MAPK signaling pathway. In all cases, BF-OXY-TAc was a more effective agent in its protective effect against the inflammatory response, as shown by its enhanced ability to reverse the inflammation-induced alteration of p-p38, p-ERK, and p-JNK protein expression compared to BF-OXY.

## 4. Discussion

OXY has attracted considerable interest from the scientific community for its pharmacological benefits, including its anti-inflammatory activities. However, therapeutic applications of OXY remain minimal due to its physicochemical characteristics and pharmacokinetic profile. OXY exhibits low aqueous solubility (≅ 0.47 mg/mL [17]). Additionally, OXY is light-sensitive. It is susceptible to oxidative discoloration and photoisomerization, especially when exposed to light or heat [16,17]. Moreover, pharmacokinetic studies revealed that OXY has poor oral bioavailability due to extensive hepatic metabolism and rapid elimination [14,15,33,34]. These drawbacks restrict the clinical utility of OXY. Several strategies have been used to improve the physicochemical properties and enhance the pharmacokinetic profile of OXY, such as the use of cyclodextrin encapsulation [17,35,36], nanostructured lipid carriers [5], a self-micro emulsifying drug delivery system [37], co-crystallization [38,39], microemulsion [40], and ester prodrugs [20].

In this study, a series of ester prodrugs derived from OXY, including OXY-TAc, OXY-TPr, and OXY-TBu, were synthesized and evaluated for their membrane-permeation ability across intestinal epithelium using a Caco-2-based model. The Caco-2 monolayer is a well-established tool for predicting the absorption of compounds, because its characteristics mimic the intestinal epithelium of humans. The P_app_ value from this assay is the pharmacokinetic parameter that directly correlates with membrane-permeation ability, i.e., a large P_app_ value corresponds to strong membrane-permeation ability. Our results demonstrated that the observed P_app_ value of synthesized ester prodrugs was about 1.5 to 3 times greater than the OXY parent compound, showing that the ester prodrug approach can enhance the ability of OXY to cross the membrane. The improvement of the epithelial transport of ester prodrugs could be due to increased lipophilicity from the other chain. Among the ester prodrugs derived from OXY, we found that OXY-TAc had the greatest capacity to travel across the intestinal membrane (P_app_ value; OXY-TAc (4.25 (±0.43) × 10^−6^ cm/s) > OXY-TPr (2.67 (±0.19) × 10^−6^ cm/s) > OXY-TBu (2.17 (±0.16) × 10^−6^ cm/s)). The values of calculated log solubility (clogS) for OXY-TAc (−5.135), OXY-TPr (−5.956), and OXY-TBu (−7.619) were also decreased compared to OXY (−2.505). As side-chain length increased, the ester prodrugs showed consistent increases in lipophilicity, with corresponding decreases in aqueous solubility [41]. We postulated that the superior membrane-permeation ability of OXY-TAc could be due to its physicochemical property. According to Lipinski’s Rule of Five, compounds with calculated log P (clogP) less than 5 are more likely to have a good absorption profile [42]. The clogP of OXY-TAc was 3.52, while the clogP values of OXY-TPr and OXY-TBu were 5.15 and 6.32, respectively. Hence, OXY-TAc could be a good candidate to develop for clinical use.

Previous reports have proposed that the anti-inflammatory effects of the parent compound OXY are strongly linked to its antioxidant activity [8,10,43]. Exposure to inflammatory inducers can lead to excessive generation of ROS in immune cells, resulting in the initiation of the inflammatory process [44]. Supplementation with OXY inhibits oxidative-stress formation by several mechanisms, e.g., direct ROS scavenging [45] and enhancement of the detoxifying capacity of cellular antioxidant enzymes [46,47,48]. This reduction in the excessive generation of ROS results in a decrease in the production/release of pro-inflammatory mediators, and eventually in the inhibition of the inflammatory process [8,10]. In this study, the bioavailable fraction of OXY-TAc collected from the Caco-2-permeability assay was evaluated for its pharmacological activities using an in vitro macrophage model. RAW264.7 macrophages are commonly used cell lines to observe the anti-inflammatory responses and underlying mechanisms of testing compounds [49,50]. LPS is an endotoxin released from Gram-negative bacteria. LPS can activate the inflammatory response of macrophages by stimulating the production of various pro-inflammatory mediators [51]. Hence, LPS-stimulated RAW264.7 cells are generally used in vitro to mimic the inflammatory milieu [49,52]. For the evaluation of the antioxidant properties of OXY-TAc, we used the DCFH-DA-based approach. A fluorogenic probe, DCFH-DA, is used to predict the oxidative status of cells. After entering cells, DCFH-DA is deacetylated by intracellular esterases to obtain non-fluorescent DCFH. In the presence of oxidative stress, DCFH undergoes oxidation reaction resulting in the formation of fluorescent product DCF, which can be detected with several fluorescence-based techniques [53]. Following LPS stimulation, intracellular oxidative stress was generated on macrophages, as determined by an increase in DCF fluorescent signals. These results were consistent with previous reports using this model. Supplementation with BF-OXY-TAc significantly suppressed the formation of oxidative stress in LPS-induced RAW264.7 cells, suggesting its antioxidant activity. Moreover, we found that the magnitude of inhibitory effect of OXY-TAc was more potent than its parent compound, OXY. Hence, we postulated that the ester prodrug approach would enhance the therapeutic effectiveness of OXY.

We further questioned whether OXY-TAc could suppress inflammatory reactions by monitoring important biomarkers, including pro-inflammatory mediators and inflammation-regulated enzymes. LPS is an inflammatory ligand that can bind to Toll-like receptor 4 (TLR4) on the cell membrane of macrophages, leading to the activation of inflammatory cascades, the release of pro-inflammatory mediators, and eventually the occurrence of inflammation [54]. IL-6 and TNF-α are principal cytokines that play an essential role in regulating inflammatory signals. The secretion of these cytokines from macrophages can lead to the stimulation of immune systems and induction of the production/release of the consequent cytokines [55,56]. Our ELISA data demonstrated that supplementation with BF-OXY-TAc significantly suppressed the release of IL-6 and TNF-α in LPS-stimulated RAW264.7 cells. In addition to pro-inflammatory cytokines, we observed the effects of OXY-TAc on the protein expression of key enzymes that regulate inflammation, including iNOS and COX-2. iNOS is one of the major enzymes responsible for the inflammatory response of macrophages. This enzyme is typically not expressed in a resting macrophage. The expression of iNOS must be induced by inflammatory stimuli (e.g., LPS) or pro-inflammatory cytokines (e.g., TNF-α, IFN-γ and IL-1). iNOS regulates the synthesis of NO from L-arginine and molecular oxygen. The NO produced from iNOS is considered an inflammatory signaling molecule that can trigger an inflammatory response [57,58]. Our results showed that pre-treatment with BF-OXY-TAc prior to LPS stimulation markedly mitigated both the expression of iNOS and the production of NO. COX-2 is the primary molecular target for inflammatory therapy. It is an inducible enzyme that regulates the formation of inflammatory mediator PGE2. Similarly to iNOS, the expression of COX-2 is maintained at a low level in resting macrophages, and can be dramatically induced in response to pro-inflammatory stimuli [59]. During the inflammation response, the excessive production of PGE2 due to induction of COX-2 exacerbates a broad range of biological events associated with inflammation, such as vasodilation and enhancement of vascular permeability [60]. Results from the western blot assay demonstrated that pre-treatment with BF-OXY-TAc significantly inhibited COX-2 expression in LPS-stimulated RAW264.7 cells. In addition, a previous report on a similar in vitro model demonstrated that the OXY parent compound could suppress the release of PGE2 from LPS-induced-RAW264.7 cells [8]. Hence, a similar inhibitory effect on the PGE2 level would be expected with BF-OXY-TAc. Moreover, we further elucidated the molecular mechanism of OXY-TAc that regulates inflammatory response. MAPK signaling cascades comprise a family of serine/threonine kinases that transduce extracellular stimuli into a wide range of biological events through a series of phosphorylations. Stimulation of these pathways by inflammatory stimuli leads to an increase in the production of the pro-inflammatory mediators and downstream signaling events that mediate the inflammatory response [4]. In this study, we observed the effects of OXY-TAc on three primary kinases of MAPK pathways, namely, p38, ERK, and JNK, by monitoring their phosphorylation status with western blot analysis. We found that OXY-TAc significantly prevented the phosphorylation of these kinases in LPS-stimulated RAW264.7 cells, suggesting that MAPK pathways contribute to the anti-inflammatory activities of OXY-TAc. Taken together, the results from the in vitro macrophage model provided the potential mechanisms of the anti-inflammatory effects of OXY-TAc; OXY-TAc suppressed the release of pro-inflammatory cytokines (IL-6 and TNF-α), a pro-inflammatory mediator (NO), and inflammatory regulating enzymes (iNOS and COX-2) via inhibition of MAPK cascades. Alongside its enhanced antioxidant activity, we found that the anti-inflammatory properties of OXY-TAc were greater than its parent compound OXY. We postulated that the increase in pharmacological activity of OXY-TAc could be due to an improvement in physicochemical property and bioavailability profile. These results strongly support the potential use of OXY-TAc for the treatment of inflammation-related disorders.

## 5. Conclusions

OXY ester prodrugs could increase the transport of OXY across Caco-2 monolayers. OXY-TAc enhances the anti-inflammatory activity of OXY on LPS-stimulated RAW264.7 macrophage cells. According to the findings of this study, BF-OXY-TAc significantly inhibited the production of ROS, NO, IL-6, and TNF-α, as well as the expression of iNOS, COX-2, and MAPK (p38, ERK, and JNK) proteins in LPS-stimulated RAW264.7 macrophages, which suggests that OXY-TAc may be one of the ester prodrugs suitable for inflammation therapy. The enhancement of activity could be derived from the higher membrane permeation, which in turn might be derived from an improvement in the lipophilicity of the OXY. We reported herein for the first time the anti-inflammatory activity of OXY-TAc, as well as the potential use of BF-OXY-TAc to assess the biochemical profile alteration of LPS-activated RAW264.7 macrophage cells.

## Figures and Tables

**Figure 1 molecules-27-03922-f001:**
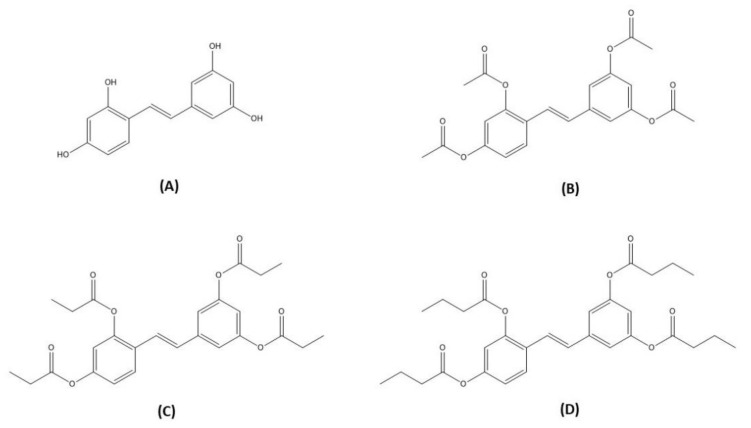
Structures of Oxyresveratrol (OXY; (**A**)), oxyresveratrol tetraacetate (OXY-Tac; (**B**)), Oxyresveratrol tetrapropionate (OXY-TPr; (**C**)), and oxyresveratrol tetrabutyrate (OXY-TBu; (**D**)).

**Figure 2 molecules-27-03922-f002:**
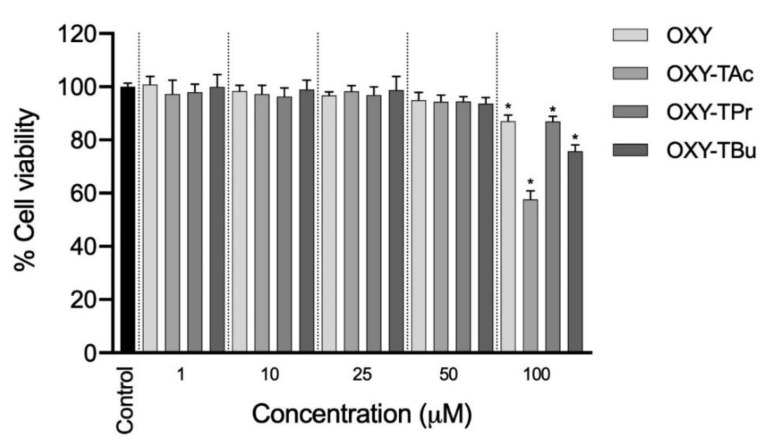
The viability of Caco-2 cells after 24 h exposure in OXY and four OXY ester prodrugs (1–100 µM). The data are expressed as the mean of four replicates ± SD. ^*^ *p* < 0.05 vs. control group.

**Figure 3 molecules-27-03922-f003:**
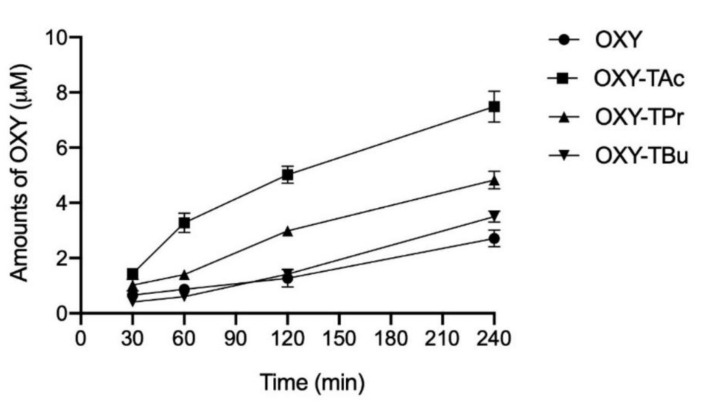
Time course of transport of OXY derived from OXY or OXY ester prodrugs across the Caco-2 cell monolayers. The concentration of each compound at selected timepoints in basolateral compartment was measured using HPLC. The data are expressed as the mean of four replicates ± SD.

**Figure 4 molecules-27-03922-f004:**
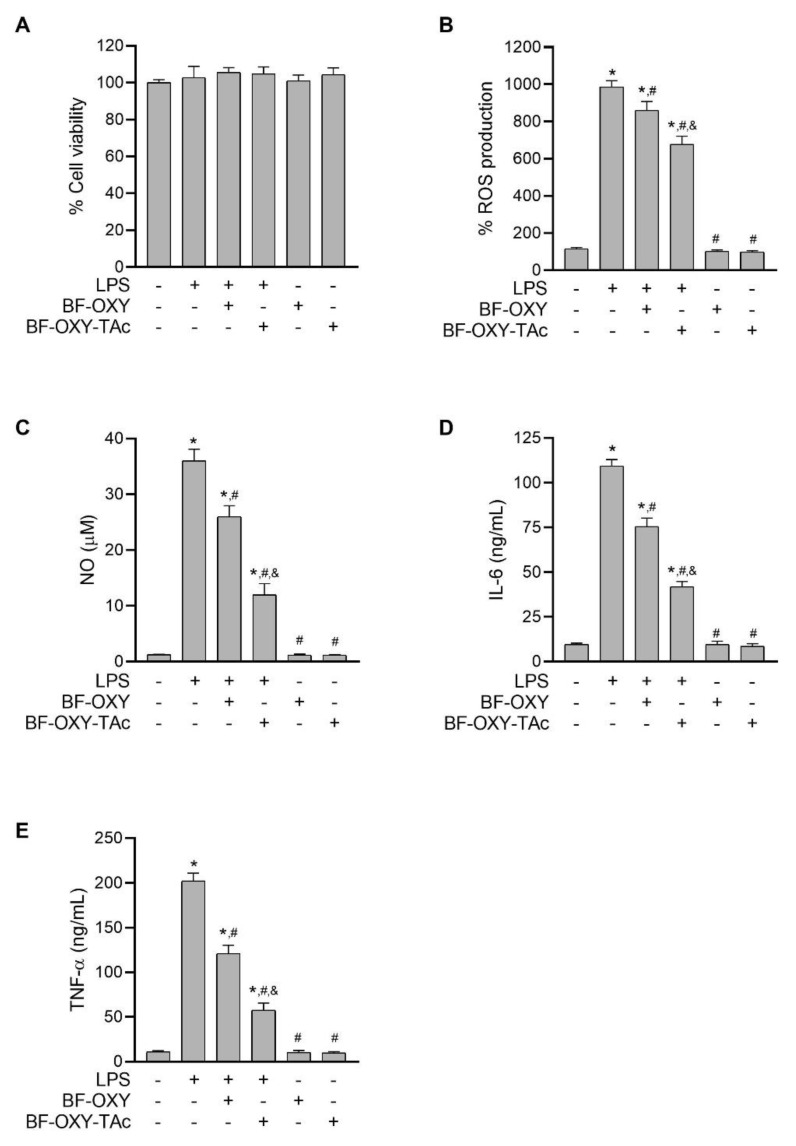
The effects of BF-OXY-TAc on cytotoxicity, ROS, NO, IL-6, and TNF-α production in LPS-stimulated RAW264.7 cells. The cells were pre-treated with BF-OXY or BF-OXY-TAc for 1 h before treatment with LPS (1 µg/mL) for 24 h. (**A**) The cell viability was analyzed with the MTT method. (**B**) The evaluation of oxidative stress was conducted with a DCFH-DA-based assay. (**C**) The measurement of NO formation was performed using the Griess reaction. (**D**,**E**) The determination of IL-6 and TNF- α was performed by ELISA. The data are expressed as the mean of four replicates ± SD. ^*^ *p* < 0.05, vs. untreated control; ^#^ *p* < 0.05, vs. LPS; ^&^ *p* < 0.05, vs. LPS and BF-OXY treated group.

**Figure 5 molecules-27-03922-f005:**
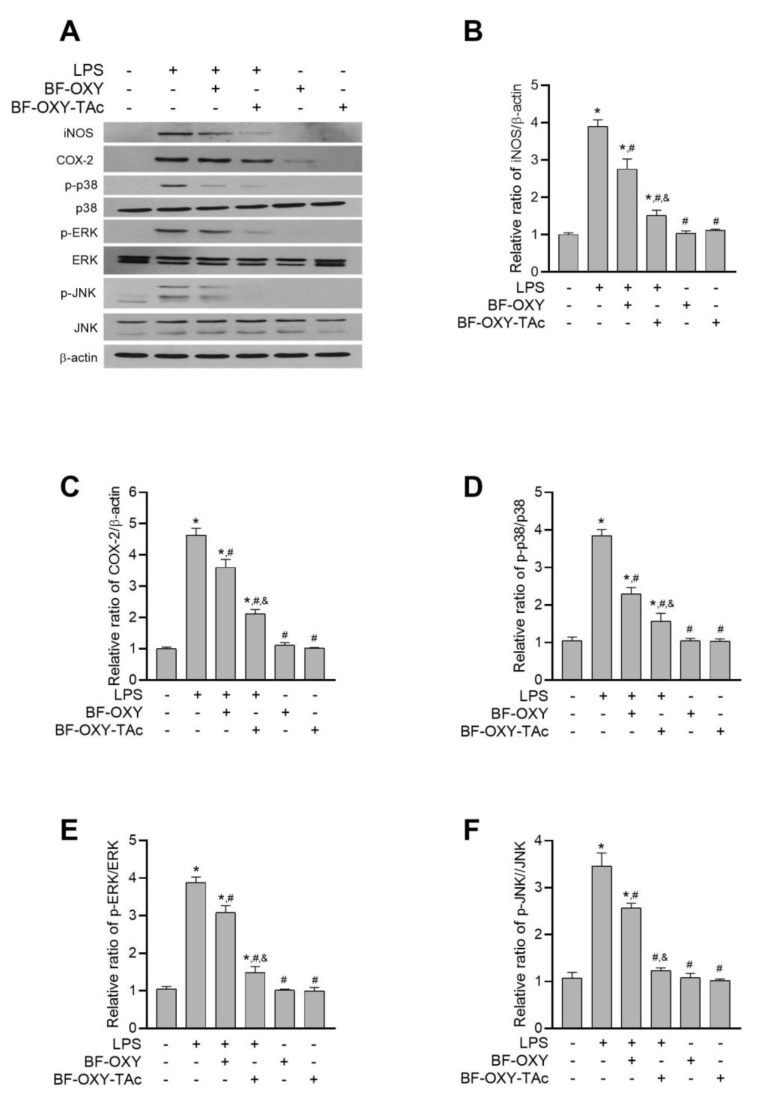
The effects of BF-OXY-TAc on iNOS and COX-2 and MAPK cascades in LPS-stimulated RAW264.7 cells. (**A**) The protein expressions of iNOS, COX-2, p38, ERK, and JNK were observed with western blot analysis. Results are representative of four independent experiments. (**B**–**F**) Band intensity of iNOS, COX-2, p38, ERK, and JNK expression was quantified. The RAW264.7 cells were pre-treated with BF-OXY or BF-OXY-TAc for 1 h before being induced with LPS (1 µg/mL) for 24 h. The data are expressed as the mean of four replicates ± SD. ^*^ *p* < 0.05, vs. untreated control; ^#^ *p* < 0.05, vs. LPS; ^&^ *p* < 0.05, vs. LPS and BF-OXY treated group.

## Data Availability

Not applicable.

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
