# Peer review of "Anti-Inflammatory Activity of Oxyresveratrol Tetraacetate, an Ester Prodrug of Oxyresveratrol, on Lipopolysaccharide-Stimulated RAW264.7 Macrophage Cells"

_molecules, 2022, doi:10.3390/molecules27123922_

Round 1
Reviewer 1 Report
The anti-inflammatory activity of oxyresveratrol (OXY) bas been widely shown to be effective against experimental inflammatory-related diseases, but rather poor pharmacokinetic profiles may limit the biologic application of the prodrug.
Thaweesest et al., synthesized a series of ester prodrugs of Oxyresveratrol (OXY), including OXY-TAc, OXY-TPr, and OXY-TBu and attentively evaluated their membrane permeabilities across a monolayer of Caco-2 cells. The study demonstrates the OXY-TAc showing greater transport ability than OXY. Also, the cell culture studies using a well-defined model, LPS-stimulated RAW264.7 macrophages, provide solid head-to-head evidence of OXY-TAc as an anti-inflammatory prodrug, better than OXY.
One of the points of improvement in the manuscript is the redundant description of the Methods in the Results section.
Reviewer 2 Report
In this manuscript, the authors presented that anti-inflammatory and anti-oxidant effects of OXY-Tac in LPS-treated RAW264.7 cells. They also explained the inhibition of p38, ERK and JNK is a major mechanistic signalling cascade for these beneficial activities. This might be quite an effective and promising reagent for anti-inflammatory drug. This report is helpful for readers, but it still has some problems as indicated below.
Comments
1. The dose of drugs used in Raw cells was not clearly indicated. The authors need to describe the concentration of the drug.
2. Are the anti-inflammatory effects of OXY-Tac dose-dependent?
3. As mentioned by the author, COX-2 is involved in PGE2 production. Is OXY-Tac involved in the production of PGE2?. The authors need to check the change in PGE2 levels by drug.
4. Authors showed a decrease in TNF-a and IL-6. Is IL-1b expression also decreased by OXY-Tac?
5. As the authors know, NF-kB is a major factor in oxidative stress and inflammatory response. The authors should show the change of NF-kB expression by OXY-Tac.
6. In line 244, fetal bovine serum should be abbreviated.
7. Increase the size of * in all graphs.
Reviewer 3 Report
The article I received for review is formally very well prepared. It meets all the criteria for good experimental work. The topic discussed in it also seems topical and interesting. Because resveratrol and its numerous derivatives are an extremely valuable research topic, and they also have numerous described useful applications. But this fact, unfortunately, significantly reduces the scientific level of this work. The leading compound used in it is well known and described in many previous works. Its effect on various biological systems is described, but of course, with the exclusion of the cell line and research direction used here. To complete the conducted biological research, this work describes the preparation of two new homologues of this basic compound (i.e., the acetyl derivative). However, this is not something special or original. Both their synthesis is carried out in a highly routine manner, and the results of biological research do not differ from expectations. Therefore, it is a typical work that expands our knowledge, but it cannot be considered in any way revealing or even innovative.
If this work will be approved for publication, I suggest making the following minor corrections in it:
- shortening a slightly too extensive introduction,
- an explanation in lines 115-116 which OXY substance was really used in the studies, whether isolated from a natural source or a commercial product,
- relatively low yields of acylation products require commentary; reactions of this type usually take place with very good yields,
- supplementary materials are prepared with extreme carelessness, as if they came from other authors; in this respect, they should be properly elaborated or removed.
The above remarks are made under the influence of subjective observations of the reviewer. These are not errors but only small mistakes, just issues to consider for correction in order to improve the final quality of the article.
Apart from the non-obligatory minor corrections mentioned above, the manuscript can be accepted for publication. However, in my opinion, it will remain a poor job of an average scientific level, with a quite good research technique.
Round 2
Reviewer 2 Report
I accept the present form of this study.